# The Effect of Belantamab Mafodotin on Primary Myeloma–Stroma Co-Cultures: Asymmetrical Mitochondrial Transfer between Myeloma Cells and Autologous Bone Marrow Stromal Cells

**DOI:** 10.3390/ijms24065303

**Published:** 2023-03-10

**Authors:** Zsolt Matula, Ferenc Uher, István Vályi-Nagy, Gábor Mikala

**Affiliations:** 1Laboratory for Experimental Cell Therapy, Central Hospital of Southern Pest, National Institute of Hematology and Infectious Diseases, 1097 Budapest, Hungary; uher.ferenc@dpckorhaz.hu; 2Department of Hematology and Stem Cell Transplantation, Central Hospital of Southern Pest, National Institute of Hematology and Infectious Diseases, 1097 Budapest, Hungary; valyi-nagy.istvan@dpckorhaz.hu (I.V.-N.); gmikala@dpckorhaz.hu (G.M.)

**Keywords:** multiple myeloma, belantamab mafodotin, mitochondrial transfer, cancer drug resistance, bone marrow mesenchymal stromal cell

## Abstract

Belantamab mafodotin (belamaf) is an afucosylated monoclonal antibody conjugated to the microtubule disrupter monomethyl auristatin-F (MMAF) that targets B cell maturation antigen (BCMA) on the surface of malignant plasma cells. Belamaf can eliminate myeloma cells (MMs) through several mechanisms. On the one hand, in addition to inhibiting BCMA-receptor signaling and cell survival, intracellularly released MMAF disrupts tubulin polymerization and causes cell cycle arrest. On the other hand, belamaf induces effector cell-mediated tumor cell lysis via antibody-dependent cellular cytotoxicity and antibody-dependent cellular phagocytosis. In our in vitro co-culture model, the consequences of the first mentioned mechanism can be investigated: belamaf binds to BCMA, reduces the proliferation and survival of MMs, and then enters the lysosomes of malignant cells, where MMAF is released. The MMAF payload causes a cell cycle arrest at the DNA damage checkpoint between the G2 and M phases, resulting in caspase-3-dependent apoptosis. Here, we show that primary MMs isolated from different patients can vary widely in terms of BCMA expression level, and inadequate expression is associated with extremely high resistance to belamaf according to our cytotoxicity assay. We also reveal that primary MMs respond to increasing concentrations of belamaf by enhancing the incorporation of mitochondria from autologous bone marrow stromal cells (BM-MSCs), and as a consequence, MMs become more resistant to belamaf in this way, which is similar to other medications we have analyzed previously in this regard, such as proteasome inhibitor carfilzomib or the BCL-2 inhibitor venetoclax. The remarkable resistance against belamaf observed in the case of certain primary myeloma cell cultures is a cause for concern and points towards the use of combination therapies to overcome the risk of antigen escape.

## 1. Introduction

Multiple myeloma is the second most common hematological malignancy worldwide and accounts for approximately 10% of all hematologic malignancies [1], with an average of 400–500 newly diagnosed patients registered in Hungary every year. With conventional therapies, the median survival is approximately 6 years, which can be extended with autologous stem cell transplantation [2]. In the past two decades, there has been a substantial breakthrough in the treatment of multiple myeloma as many new classes of drugs have been introduced for clinical care; the approval and routine clinical use of immunomodulatory drugs (IMiDs) and proteasome inhibitors (PIs), followed by the availability of monoclonal antibodies (mAbs), have been fundamental breakthroughs in improving survival outcomes in patients. Nevertheless, multiple myeloma remains a largely incurable malignancy [3,4]. Based on the results of a study involving 14 academic centers in the US, the median overall survival (OS) of patients refractory to anti-CD38 mAb was only 8.6 months. The median OS was 11.2 months for patients not simultaneously refractory to an IMiD and a PI, but only 5.6 months for patients who were refractory to anti-CD38 mAb, two proteasome inhibitors, and two IMiDs, showing the dismal chances of survival for these patients [5]. However, it is encouraging that the therapeutic options have been greatly expanded in recent years, and the incorporation of further new agents into routine clinical practice will hopefully significantly improve the chances of survival of these multi-refractory patients.

New approaches such as chimeric antigen receptor (CAR) T lymphocytes, bispecific antibodies, and antibody–drug conjugates (ADCs) can significantly improve outcomes for multi-refractory patients not responding to standard therapies, and these approaches represent a generational paradigm shift in the treatment of multiple myeloma [6]. B-cell maturation antigen is one of those antigens expressed on the surface of plasma cells that can be targeted by these new approaches [7]. BCMA is essential for the proliferation and survival of plasma cells and is expressed at a much higher level in the surface of myeloma cells than in the case of other cell types, minimizing the off-target effect of BCMA targeting antibody–drug conjugates [8]. In August 2020, the Food and Drug Administration granted accelerated approval to belantamab mafodotin (BLENREP; GlaxoSmithKline), a BCMA-targeted antibody–drug conjugate for the treatment of patients with relapsed or refractory multiple myeloma [9]. Belamaf treatment can be administered to patients who have previously received at least four therapies including an anti-CD38 monoclonal antibody, an IMiD, and a proteasome inhibitor [10]. The DREAMM (Driving Excellence in Approaches to Multiple Myeloma) clinical trials initially demonstrated that belamaf treatment results in a promising overall response rate and progression-free survival even when employed as a monotherapy [11,12]. Subsequent DREAMM studies demonstrated deep and durable responses in the heavily pretreated population [13,14,15], and several ongoing studies are still investigating the effectiveness of belamaf as a monotherapy (NTC04162210, NTC04398745, NTC04398680, NTC05064358) or in combination with other medications (NTC03848845, NTC04126200, NTC03544281, NTC04246047, NTC04484623, NTC04091126, NTC03715478) [16,17,18,19,20,21,22,23].

Belantamab mafodotin specifically binds BCMA and eliminates multiple myeloma cells by a multimodal mechanism of action including the inhibition of BCMA receptor signaling and microtubule polymerization, the induction of antibody-dependent cellular cytotoxicity (ADCC), and antibody-dependent cellular phagocytosis (ADCP) [24]. Moreover, the release of markers characteristic of immunogenic cell death potentially leads to an adaptive immune response and immunologic memory [25]. An important difference between belantamab and other therapeutic antibodies such as daratumumab and isatuximab is based on the structure of the Fc region. For daratumumab and isatuximab, the majority of the N-linked biantennary complex-type oligosaccharides are core fucosylated, whereas belantamab is afucosylated, which provides an improved binding to FcγRIIIa on the surface of natural killer cells and enhances ADCC [26]. The in vitro mechanism of action in the co-culture of bone marrow stromal cells (BM-MSCs) and MMs relies on two processes. On the one hand, its binding to the BCMA receptor inhibits BCMA signaling and reduces the proliferation and survival of MMs. On the other hand, belamaf is internalized into the cell and transported to the lysosomes, where monomethyl auristatin-F is released through proteolytic cleavage. MMAF disrupts the intracellular microtubule network, causing cell cycle arrest in the G2/M phase checkpoint and resulting in caspase 3-dependent apoptosis [27].

Here, we show that, in the case of primary myeloma cell cultures established from bone marrow aspirates of MM patients, the BCMA expression of the malignant plasma cells derived from different donors is dissimilar, as their positivity varies between 31 and 96%, according to the flow cytometric analyses. To determine the drug sensitivity of MMs in either monoculture or their co-culture with autologous BM-MSCs, we selected three primary myeloma cell cultures with relatively low, medium, or high BCMA expression. Depending on BCMA expression level, very different cytotoxicity curve patterns were measured with high-content screening analysis by applying the same belamaf concentrations in our in vitro viability assays. In the vast majority of cases, although not for all concentrations of belamaf, we observed that the survival of MMs was improved when cultured together with BM-MSCs, rather than without the stromal cells. Therefore, we performed mitochondrial transfer assays as we previously demonstrated that bone marrow stromal cells led to the drug resistance of MMs via mitochondrial transfer (MT) in their co-culture in response to in vitro carfilzomib, venetoclax, or Na-valproate treatment [28]. We found that the MT from the stromal cells toward the MMs increases proportionally with increasing belamaf concentrations. In contrast, in the opposite direction, from the MMs to the stromal cells, the intensity of MT decreased proportionally with increasing concentrations of belamaf. We found no significant difference between BCMA^high^ and BCMA^low^ myeloma cells regarding the alteration in MT intensity in the presence of belamaf: the BCMA^low^ myeloma cells acquired a significantly higher amount of MSC-derived mitochondria compared to the untreated co-cultures, similar to the BCMA^high^ MMs.

## 2. Results

### 2.1. BCMA Positivity of Myeloma Cells in Primary BM-MSC–MM Co-Cultures Derived from Different Multiple Myeloma Patients

First, we determined the BCMA positivity of primary MMs obtained from ten patients with intramedullary myeloma. Primary BM-MSC–MM co-cultures were established by in vitro culturing the mononuclear cells previously separated from bone marrow aspirates by density gradient centrifugation using Ficoll-Paque. The myeloma cells were removed from the adherent stromal cells by washing immediately before antibody staining (anti-BCMA or the corresponding isotype control antibody) and flow cytometric measurement to avoid phenotypic changes of the MMs that could be caused by the absence of the stromal cells. The BCMA expression of the MMs was highly variable, as the frequency of BCMA-positive MMs ranged between 31 and 96% in our primary BM-MSC–MM co-cultures (Figure 1A). These values were highly similar across the three independent measurements, as shown by the low standard deviations (Figure 1C). The age, gender, primary genetic alteration and Ig isotype of the malignant plasma cells, the disease and treatment stage, and drug resistance of the patients involved in this study are detailed in Appendix A.

### 2.2. Cytotoxic Effects of Belantamab Mafodotin on Primary BM-MSC or Myeloma Monocultures, or Their Co-Cultures

To determine the cytotoxicity of belantamab mafodotin on primary myeloma cell cultures, first, we analyzed the effect of belamaf on the viability of MMs and BM-MSCs in their monocultures. To reveal the presumable protective effect of BM-MSCs on myeloma cells’ survival in the presence of increasing concentrations of the antibody–drug conjugate, we measured the viability of malignant plasma cells in the BM-MSC–MM co-cultures as well using the concentration range of belamaf between 0.1 and 1000 µg/mL. Three distinct myeloma cell cultures were selected for these experiments with different BCMA expressions. A low (31% positivity), an intermediate (58% positivity), and a high (96% positivity) BCMA-expressing cell culture were analyzed using a high-content screening method after 72 h of incubation. Our results show that the BCMA expression level strongly correlates with drug toxicity on myeloma cells in monocultures and co-cultures (Figure 1B,C).

Based on our results, it can also be concluded that the viability of malignant plasma cells is higher when cultured together with autologous stromal cells than in monocultures; thus, BM-MSCs somehow support the survival of malignant plasma cells and delay their apoptosis in the presence of belamaf. However, the differences observed between the mono- and co-cultures in terms of the viability of MMs were not significant at each drug concentration, as shown in Figure 1B. Finally, it is also evident from the viability assay that the stromal cells are also eradicated by the higher concentrations of belamaf, although not to the same extent as the malignant plasma cells. Although BCMA is not expressed on the surface of BM-MSCs, the microtubule disrupter monomethyl auristatin-F certainly penetrates the stromal cells and induces programmed cell death. This mechanism most likely occurs due to pinocytosis.

### 2.3. Mitochondrial Transfer between BM-MSCs and Malignant Plasma Cells in the Presence of Higher Doses of Belantamab Mafodotin

Previously, we revealed that conventional anti-myeloma medicines such as the proteasome inhibitor carfilzomib, the BCL-2 inhibitor venetoclax, or the HDAC inhibitor Na-valproate induce an intensive, bidirectional transfer of functional mitochondria between bone marrow stromal cells and MMs, which provides remarkable resistance to these pharmaceuticals for the malignant plasma cells. To uncover whether belamaf induces a similar process and enhances bidirectional MT between the stromal cells and MMs, we monitored the change in MT in primary BM-MSC–MM co-cultures utilizing high concentrations of belamaf (500 µg/mL, 1000 µg/mL). Colcemid was also tested as a control reagent, which similarly acts like the MMAF payload of belamaf and inhibits tubulin polymerization.

The functional mitochondria derived from BM-MSCs previously labeled by MitoTracker Red FM dye were incorporated by MMs (identified by their positivity for CD38) even without belamaf treatment after 48 h of co-culture, but 500 µg/mL and 1000 µg/mL belamaf significantly increased the MT in a dose-dependent manner by 22% and 33%, respectively, compared to the untreated cell cultures (Figure 1D). Increasing the concentration of belamaf led to the elevated apoptosis of MMs, as shown by the cytotoxicity assays, but MMs that survived incorporated an increasing amount of BM-MSC-derived mitochondria in response, resulting in elevated resistance to the cytotoxic and pro-apoptotic effect of belamaf. The transmission of mitochondria from the malignant plasma cells toward the BM-MSCs in the presence of belamaf was also investigated by measuring the positivity of CD146^+^ BM-MSCs for MM cell-derived mitochondria after 48 h of co-culture. Importantly, while belamaf boosted the MT from BM-MSCs to MM cells, 500 µg/mL and 1000 µg/mL belamaf significantly reduced the MT in the opposite direction in a dose-dependent manner by 11% and 24%, respectively (Figure 1E). The effect of colcemid was almost identical to belamaf regarding the alteration of MT intensity as 1 µM colcemid significantly enhanced the MT from BM-MSCs to MMs (by 44% on average), but partially blocked the organelle transfer from MMs towards the BM-MSCs (by 14.5% on average). These results are only partially comparable to those obtained with the aforementioned medicines (carfilzomib, venetoclax, Na-valproate), where the number of transferred mitochondria was greatly increased by these drugs both from BM-MSCs to MMs and from MMs to BM-MSCs. Interestingly, we found no correlation between the drug sensitivity of the malignant plasma cells, which correlates with their BCMA positivity, and the increase in MT under the influence of belamaf; although the viability of the three cell cultures was dissimilar at a defined belamaf concentration depending on the BCMA expression of the MMs, the percentage increase in the MT was almost identical in all three cultures compared to the untreated control cultures. This highlights that myeloma cells respond to even mild mitochondrial damage by intensive mitochondrial incorporation derived from the stromal cells to counteract the mitochondrial dysfunction and promote survival.

## 3. Discussion

Although multiple myeloma accounts for approximately 10% of hematologic malignancies and has the second highest incidence, it is still generally considered an incurable disease. However, with the introduction of novel therapies into the standard of care in the recent decade, such as second-generation proteasome inhibitors (carfilzomib, ixazomib), third-generation immunomodulatory drugs (pomalidomide), HDAC inhibitors (panobinostat), monoclonal antibodies (daratumumab, isatuximab, elotuzumab), and the BCL-2 inhibitor venetoclax, approximately 90% of myeloma patients have a considerable chance of reaching complete remission and measurable residual disease negativity [4,29,30]. Moreover, the latest therapies including bispecific antibodies, antibody–drug conjugates, and chimeric antigen receptor T cells show promising efficacy even for multi-refractory patients with high-risk features and may help to achieve and maintain deep and highly durable responses [6,31,32,33].

The antibody–drug conjugate belantamab mafodotin was first approved in 2020 by the FDA for relapsed/refractory multiple myeloma (RRMM) patients as a monotherapy treatment [9]. The original authorization was applicable for those myeloma patients who have received at least four prior therapies including an anti-CD38 monoclonal antibody, a proteasome inhibitor, and an immunomodulatory drug. Belamaf is an afucosylated IgG1 bioconjugated to the microtubule disrupter monomethyl auristatin-F (MMAF) via a non-cleavable maleimide linker. It targets B-Cell Maturation Antigen and eliminates MMs by several mechanisms: (i) the inhibition of the proliferation and survival of MM cells via prohibiting BCMA receptor signaling; (ii) the disturbance of microtubule polymerization and thus inducing cell cycle arrest in the G2/M phase checkpoint, and a caspase 3-dependent apoptosis; (iii) the induction of antibody-dependent cellular cytotoxicity (ADCC); (iv) the induction of antibody-dependent cellular phagocytosis (ADCP); and (v) causes the release of immunogenic cell death markers and possibly promotes an adaptive immune response [25,27,34,35]. Currently, a total of 37 clinical trials, including the studies of the DREAMM clinical development program, are in progress worldwide and investigating the BCMA-targeted antibody–drug conjugate belamaf therapy in multiple myeloma either as a third- or fourth-line monotherapy or as a first-, second-, third-, or fourth-line combination therapy using belamaf together with different anti-cancer drugs or monoclonal antibodies [16,17,18,19,20,21,22,23,36,37].

We started our experiments by determining the BCMA protein expression level of primary MMs isolated from the bone marrow of multiple myeloma patients by flow cytometry. BCMA is a cell surface receptor belonging to the tumor necrosis factor receptor family, which shows high expression in both mature B-lymphocytes and plasma cells, but its expression is significantly increased on the surface of MMs compared to normal plasma cells. For this reason, many clinical trials are currently using the BCMA-targeted treatment strategy with antibody–drug conjugates or CAR-T cell products [7,34,38,39,40]. However, several studies have reported contrary data regarding the BCMA expression of primary MMs isolated from the bone marrow of myeloma patients. Based on these investigations, the BCMA expression seems to vary greatly from patient to patient, and it is even possible that only ~25% of neoplastic plasma cells show positivity for the BCMA receptor based on the flow cytometric measurements [41,42,43,44,45]. Of course, this does not exclude the possibility of detecting BCMA expression at the mRNA level with more sensitive laboratory techniques (e.g., mRNA microarray, real-time qRT-PCR, or RNA-seq). Still, for the appropriate therapeutic efficacy of an antibody–drug conjugate, there must be a suitable antigen density on the target cells’ surface. A recently published important study also highlighted that the biallelic loss of BCMA is one of the mechanisms of resistance to anti-BCMA CAR T-cell therapy with idecabtagene vicleucel [46]. Our results also show that in primary BM-MSC–MM co-cultures established from bone marrow aspirates of myeloma patients, the BCMA expression of neoplastic plasma cells varies greatly between patients (~31–96%) according to the flow cytometric measurements. As a result of clonal competition with alternating dominance in multiple myeloma, BCMA-negative myeloma subclones or myeloma subclones with low BCMA expression can develop alternative pathways to survive without BCMA. Of course, there is a chance that these subclones only lost their BCMA expression under in vitro conditions or became dominant in the cell culture during in vitro cultivation, but this is not at all certain to be the case. On the one hand, 70–80% of our primary cell cultures showed robust BCMA expression according to flow cytometric analysis, and this level did not decrease during culture, which was only a few passages from the isolation until the end of the three repeated measurements. Similar to the myeloma cell lines, much more intensive and long-term cultivation would be required for a subclone to lose its BCMA positivity and outgrow the BCMA-positive cell population. On the other hand, even if the reason for low BCMA subclones becoming dominant is due to cell cultivation, this could also happen in the bone marrow of the patients due to the high selection pressure caused by drug treatments, especially since it is a heavily pretreated patient group.

In our subsequent experiments, we selected three different myeloma cell cultures with high (96%), medium (58%), and low (31%) BCMA expression for testing the cytotoxicity of belamaf in the case of myeloma and BM-MSC monocultures, or BM-MSC–MM co-cultures at the concentration range of 0.1–1000 µg/mL using the HCS method. Our results confirmed that the expression level of BCMA strongly correlates with drug toxicity exerted on MMs both in monocultures and co-cultures. Based on our results, it can also be concluded that the viability of malignant plasma cells is higher when cultured together with autologous BM-MSCs than alone; thus, BM-MSCs somehow support the survival and delay the apoptosis of malignant plasma cells in the presence of belamaf. It should be noted that the observed differences between mono- and co-cultures regarding the viability of MMs were not significant for each drug concentration. These results are consistent with our previous findings, where the resistance of MMs increased when they were cultured together with autologous BM-MSCs in the presence of either the proteasome inhibitor carfilzomib, the BCL2 inhibitor venetoclax, or the HDAC inhibitor Na-valproate [28].

Finally, our mitochondrial transfer assay revealed how BM-MSCs can support the survival and delay the apoptosis of malignant plasma cells in the presence of belamaf. Previously we demonstrated that the transfer of BM-MSC-derived mitochondria into the MMs was greatly increased by carfilzomib, venetoclax, and Na-valproate in a dose-dependent fashion. The results obtained in the case of belamaf were highly similar: compared to untreated co-cultures, malignant plasma cells received even more functional mitochondria from BM-MSCs due to the treatment, which promoted their resistance to belamaf. However, importantly, we obtained different results regarding the MT for the opposite direction, from MMs towards the stromal cells. In the presence of carfilzomib, venetoclax, and Na-valproate, the MT from MMs towards the BM-MSCs was also increased compared to the untreated co-cultures, promoting MMs to discard their damaged or dysfunctional mitochondria, resulting in decreased ROS levels. However, in the presence of belamaf, MT was inhibited in this direction, and additionally, the same results were obtained using colcemid, which similarly to the MMAF payload of belamaf, inhibits tubulin polymerization. Additionally, we previously observed a similar trend using cytochalasin D, which inhibits actin polymerization; the MT was increased from BM-MSCs to MMs, but prohibited in the opposite direction. One hypothesis explaining this phenomenon might be that the inhibition of actin or tubulin polymerization hinders the transfer of mitochondria through the tunneling nanotubes (TNTs) of plasma cell origin, since it has already been demonstrated that MT through TNTs requires both actin and tubulin polymerization [47]. However, when TNT formation and organelle transfer are inhibited through these structures between MMs and BM-MSCs, either by cytochalasin D, colcemid, or belamaf, MMs respond with very close adherence to the stromal cells and MT continues from BM-MSCs to MM cells even more intensively through myeloma cell-derived cell projections after the partial cell membrane fusion of the two cell types. We previously demonstrated the rising dominance of this mechanism in the presence of cytochalasin D, where the number of TNTs between BM-MSCs and MMs was radically reduced, but the MSC-derived mitochondrial incorporation by MMs was significantly increased. In light of the current results, the same process occurs in the presence of belamaf in terms of mitochondrial transfer.

## 4. Materials and Methods

### 4.1. Cell Isolation and Culture

Experiments using primary cells from patients with intramedullary myeloma were approved by the Ethics and Scientific Committee of the Central Hospital of Southern Pest—National Institute of Hematology and Infectious Diseases (OGYÉI/50268-8/2017). Bone marrow aspirates were collected by sternal bone marrow puncture after patients provided written informed consent. Bone marrow mononuclear cells (BM-MNCs) were isolated by density gradient centrifugation using Ficoll-Paque PLUS (GE Healthcare Bio-Sciences, Pittsburgh, PA, USA) according to the manufacturer’s instructions. The BM-MNCs were then cultured for 3 days in DMEM/F12 growth medium supplemented by 10% *v*/*v* FBS, 2 mM l-glutamine, 100 IU/mL of penicillin, and 100 µg/mL of streptomycin. Medium and all supplements were purchased from Thermo Fisher Scientific (Waltham, MA, USA). After 3 days, the growth medium was changed, and the cells were cultured for at least four weeks. Cell cultures containing only stromal cells and intensively proliferating malignant plasma cells were trypsinized, washed with PBS buffer (purchased from Thermo Fisher Scientific), and finally cryopreserved or prepared for subsequent experiments. BM-MSCs intended for experimental purposes were rid of the MMs by repetitive thorough washing with culture media and passaging. If needed for pure separation, stromal cells were labeled with anti-CD146 Alexa Fluor 488 antibody (BioLegend, San Diego, CA, USA) and sorted by the FACSAria flow cytometer (BD Biosciences, Franklin Lakes, NJ, USA). Homogeneous primary MMs were obtained from BM-MSC–MM co-cultures by washing off the MMs attached to the adherent BM-MSC monolayer with the growth medium. Only completely separated homogeneous BM-MSC or MMs cultures were used for further experiments.

### 4.2. Flow Cytometry

To determine the BCMA positivity of primary MMs, co-cultures were washed with PBS buffer, trypsinized, and resuspended in 0.5% BSA/PBS buffer (buffers and trypsin were purchased from Thermo Fisher Scientific). Anti-CD38 Alexa Fluor 488 (BioLegend) was used to distinguish MMs from BM-MSCs. The cells were incubated with the fluorescently labeled monoclonal antibody in 0.5% BSA/PBS for 30 min at 37 °C, washed twice with 0.5% BSA/PBS buffer (centrifuged at 300× *g* for 10 min), and finally resuspended in fresh buffer. Then, the BCMA positivity of MMs was determined by labeling the MMs with anti-CD269 (BCMA) PE-conjugated antibody (Miltenyi Biotech, Bergisch Gladbach, Germany) or the corresponding isotype control antibody according to the manufacturer’s instructions: cells were washed with 0.5% BSA/PBS buffer, resuspended in fresh buffer, and incubated with the antibody for 10 min in the dark at 4 °C. After washing with buffer, the cells were resuspended and analyzed with a MACSQuant Analyzer 10 Flow Cytometer; the data were analyzed with MACSQuantify Software 2.13 (Miltenyi Biotech).

### 4.3. In Vitro Cytotoxicity Assay

The cytotoxic effects of belantamab mafodotin on separate BM-MSC or myeloma cultures, or BM-MSC–MM co-cultures were determined utilizing a high-content screening (HCS) method using the ImageXpress Pico Automated Cell Imaging System (Molecular Devices, San Jose, CA, USA). In the case of monocultures, 1 × 10^3^ BM-MSCs/well or 1 × 10^4^ MMs/well, while in the case of co-cultures, 1 × 10^3^ BM-MSCs + 1 × 10^4^ MMs/well were seeded on 96-well TC plates (Eppendorf, Hamburg, Germany). The cells were incubated in the presence of Blenrep (stock solution: solution for infusion—50 mg/mL), employing a concentration range of 0.1–1000 µg/mL for 72 h at 37 °C in a CO_2_ incubator. After incubation, Hoechst 33342 dye was added to each well in a final concentration of 100 ng/mL for 1 h to discriminate stromal cells and malignant plasma cells based on their nucleus size. Finally, propidium iodide was added to each well at a final concentration of 1 µg/mL. The viability of belamaf-treated or untreated cell cultures was evaluated using the CellReporterX-press Image Acquisition and Analysis Software (Molecular Devices).

### 4.4. Mitochondrial Transfer Assay

Separated BM-MSCs or MMs were labeled with MitoTracker Red FM dye (Thermo Fisher Scientific) at a final concentration of 200 nM in 1X HBSS buffer at 37 °C for 15 min. Next, cells were washed with supplemented DMEM/F12 growth medium three times. BM-MSCs were seeded in a 24-well plate (Eppendorf) at a density of 2.5 × 104 cells/well, while MMs were transferred into T75 flasks (Eppendorf). Stained MMs or BM-MSCs were further cultured for 72 h at 37 °C in a CO_2_ incubator then the stained cells were washed again with DMEM/F12 growth medium. Co-cultures were established using a 1:10 BM-MSC (2.5 × 104 cells/well): MM cell (2.5 × 105 cells/well) ratio. MitoTracker-labeled BM-MSCs and unlabeled MMs or MitoTracker-labeled MMs and unlabeled BM-MSCs were seeded on 24-well plates, and the cells were incubated for 48 h in the presence or absence of belantamab mafodotin (500 µg/mL or 1000 µg/mL final concentration). After culturing the cells together for 48 h, the cells were trypsinized, washed, and incubated with fluorescently labeled anti-CD146 Alexa Fluor 488 or anti-CD38 Alexa Fluor 488 monoclonal antibody (BioLegend) to distinguish MitoTracker unlabeled BM-MSCs/MMs and the other cell type previously stained with the MitoTracker Red FM dye. MitoTracker red fluorescence was analyzed by a MACSQuant Analyzer 10 Flow Cytometer (Miltenyi Biotec).

### 4.5. Statistical Evaluation

The data are presented as the mean of three repeated experiments of biological parallels ± SD. Statistical significance was tested using paired Student’s *t*-tests, and *p* < 0.05 was considered to indicate a significant difference.

## 5. Conclusions

In summary, we can conclude that the cytotoxic effect of belantamab mafodotin on primary myeloma cells varies depending on the BCMA protein level of the malignant plasma cells. However, when cultured with autologous BM-MSCs, the resistance of MMs to belamaf increases, although not significantly at each concentration. In our co-culture model, the resistance and survival of MMs were significantly enhanced by the acquisition of functional mitochondria derived from BM-MSCs in answer to belamaf treatment, similar to the proteasome inhibitor carfilzomib, the BCL-2 inhibitor venetoclax, or the HDAC inhibitor Na-valproate. As we previously demonstrated, these medicines ultimately damage the mitochondria of MMs, to which the response involves the recruitment of functional mitochondria from the stromal cells. Here, we show that the same resistance mechanism occurs in vitro as a result of belamaf treatment. It is important to emphasize that the BCMA positivity of some myeloma subclones can be radically reduced spontaneously or as a possible result of long-term drug treatment, which can be a resistance mechanism against BCMA-targeted therapies. These subclones may not only be detectable in primary cell cultures, but probably also in the patient’s bone marrow. Therefore, it is definitely worth giving priority to combined therapies over belamaf monotherapy, as this way the chance of antigen escape can be significantly reduced, and hopefully, we may get closer to challenging the dogma that multiple myeloma is an incurable disease.

## Figures and Tables

**Figure 1 ijms-24-05303-f001:**
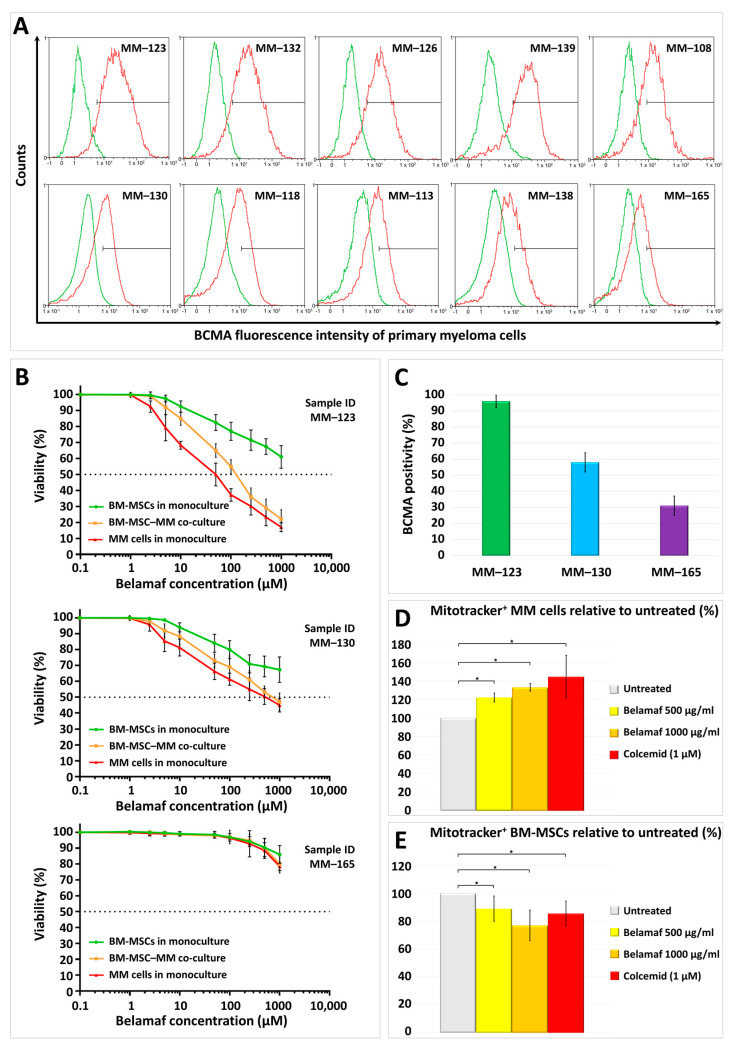
BCMA positivity of primary myeloma cells, the cytotoxic effect of belantamab mafodotin on BM-MSC–myeloma cell cultures, and the consequence of in vitro exposing the primary myeloma cell cultures to belantamab mafodotin regarding the bidirectional mitochondrial transfer between the malignant plasma cells and the bone marrow stromal cells. (**A**) CD38-positive primary myeloma cells from the BM-MSC–MM co-cultures were labeled with PE-conjugated anti-BCMA antibody or the corresponding isotype control and analyzed by flow cytometry; (**B**) the cytotoxic effect of increasing belantamab mafodotin concentrations (0.1–1000 µg/mL) was determined in myeloma (red line) and BM-MSC (green line) monocultures or the case of myeloma cells in BM-MSC–MM co-cultures (orange line); (**C**) the BCMA positivity of the three selected myeloma cell cultures for mitochondrial transfer assay with a high, medium, and low BCMA expression. Each column shows the average of three independent measurements; (**D**) the effect of belantamab mafodotin on mitochondrial transfer from BM-MSCs to MMs after 48 h of co-culture. BM-MSCs were labeled with Mitotracker Red FM and then cultured together with MMs in the presence or absence of belamaf and colcemid. BM-MSC-derived mitochondria^+^ MMs were analyzed by flow cytometry within the CD38^+^ myeloma cell population; (**E**) the effect of belantamab mafodotin on mitochondrial transfer from MMs to BM-MSCs after 48 h of co-culture. MMs were labeled with Mitotracker Red FM and then cultured together with BM-MSCs in the presence or absence of belamaf and colcemid. MM-derived mitochondria^+^ BM-MSCs were analyzed by flow cytometry within the CD146^+^ stromal cell population.

## Data Availability

All data analyzed during this study are included in this manuscript and the Appendix A.

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
