# Peer review of "The Effect of Belantamab Mafodotin on Primary Myeloma–Stroma Co-Cultures: Asymmetrical Mitochondrial Transfer between Myeloma Cells and Autologous Bone Marrow Stromal Cells"

_ijms, 2023, doi:10.3390/ijms24065303_

Round 1
Reviewer 1 Report
Congratulations for this nice work.
1. Does separation of MM cells from the adherent stromal cells affect the level of BCMA expression?
2. What could be the consequence of separating MM cells from the adherent stromal cells with regard to MM survival?
3. What was the disease and treatment stage of MM patients from whom the bone marrow aspirate sample were used (which type of treatment they received?)?
Author Response
Dear reviewer
We are grateful and appreciate you taking the time to review our manuscript. Based on your questions and the suggestions of reviewer 2, we carried out a thorough revision of the manuscript. We believe we answered all the questions and complied with the proposals. According to the opinion of reviewer 2, moderate English changes were required; therefore, we used MDPI’s English editing service, so the manuscript was checked by a native English-speaking editor (English Editing ID: 61401). All the modifications were accepted, but our changes are still visible by the “Track change” function.
You can find your questions and the corresponding answers below.
Sincerely yours,
Zsolt Matula
Referee report
Congratulations for this nice work.
- Does separation of MM cells from the adherent stromal cells affect the level of BCMA expression?
It is a good question, and the answer is definite yes. In most cases, the viability of the separated myeloma cells begins to decrease in a few weeks because the cells stop proliferating and undergo apoptosis. In parallel, the BCMA expression of the surviving myeloma cells is strongly reduced. This is why the BCMA positivity was examined immediately after washing off myeloma cells from the stromal cell layer in our experiments. In the revised manuscript, on page 3 (line 132), we indicate that antibody staining and flow cytometric measurements were performed immediately after separation to avoid phenotypic changes that could be caused by the absence of the stromal cells:
„ The myeloma cells were removed from the adherent stromal cells by washing immediately before antibody staining (anti-BCMA or the corresponding isotype control antibody) and flow cytometric measurement to avoid phenotypic changes of the MMs that could be caused by the absence of the stromal cells. MMs separated from the adherent stromal cells were labeled with anti-BCMA mAb or the corresponding isotype control antibody and analyzed by flow cytometry.” (Page 3, line 132)
Additionally, we note in the revised manuscript that BCMA levels were highly similar across the independent measurements:
“These values were highly similar across the three independent measurements, as shown by the low standard deviations (Figure 1C).” (page 3, line 140)
For the same reason, to establish the co-cultures for the cytotoxicity and mitochondrial transfer assays, the unlabeled or MitoTracker-labeled homogeneous myeloma cells were added to the plasma cell-free autologous stromal cell cultures as soon as possible in the presence or absence of belamaf/colcemid. For experimental purposes, on the one hand, homogeneous, plasma cell-free BM-MSC cultures were maintained, and on the other hand, BM-MSC–MM co-cultures were used as a source of myeloma cells. Myeloma cells alone, without BM-MSCs, were not cultured.
- What could be the consequence of separating MM cells from the adherent stromal cells with regard to MM survival?
As mentioned in the previous point, in the case of the vast majority of primary cultures, the viability of the separated myeloma cells decreases in a few weeks as the MMs stop proliferating and undergo apoptosis without the BM-MSCs. We have only 3-4 myeloma cultures out of the 80 cultures with excellent growth potential, where the separated myeloma cells proliferate intensively for several months without the stromal cells. But these cell cultures are more likely to be considered cell lines, so we do not even use them. Interestingly, we established 248 primary cell cultures, but only 80 have excellent growth potential regarding the myeloma cells. In the rest of the cell cultures, only a few non-dividing myeloma cells are detectable, or only BM-MSCs survived.
- What was the disease and treatment stage of MM patients from whom the bone marrow aspirate sample were used (which type of treatment they received?)?
In response to this question, we created a table summarizing the patients' gender, age at the time of sternal bone marrow puncture, Ig isotype and primary genetic alteration of the malignant plasma cells, medicines used during treatment, and the drug to which the patient became resistant. We inserted this table into the Supplementary Materials, and we refer to Table S1 in the revised manuscript:
„The age, gender, primary genetic alteration and Ig isotype of the malignant plasma cells, the disease and treatment stage, and drug resistance of the patients involved in this study are detailed in Supplementary Table S1.” (page 3, line 141)

Reviewer 2 Report
This manuscript examines the variability of BCMA expression on the surface of MM patient cells (by establishing cell cultures from 12 primary patients) and its relationship to the ability of a BCMA targeting agent, belantamab mafodotin (belamaf) to effectively kill MM cells either in monoculture or co-cultured with bone marrow stromal cells (BM-MSCs). In an independent series of experiments, they clearly show that mitochondrial transfer (MT) from BM-MSC to MM cells increases with increasing drug concentration. In contrast, MT from MM cells to BM-MSCs decreased with belamaf treatment.
The manuscript is well-written, clear, concise and the data are nicely presented. The manuscript will be of general interest to the MM research community, owing to the fact that belamaf is being used in a number of clinical trials as both monotherapy and in combination with other drugs.
Concepts and small edits to consider including or clarifying:
1) Since these cells are from primary patients, the drug resistance is not truly acquired resistance due to continuous drug treatment as in patients being treated. It would be useful if this is considered throughout the manuscript. For instance, in the abstract, consider replacing “MMs acquire noticeable drug resistance in this way” with “MMs are more resistant to belamaf”.
2) Page 2, line 94 insert “is” between mab and based
3) Page 3, line 117: replace “serve” with “led to”
4) Page3, line 119: replace “ growing” with “increasing”
5) Page 5, line 196: replace “conducted” with “led”
6) Page 6, line 221: change myeloma multiplex to multiple myeloma ?
7) On page 7, line 315. Rather than say our explanation for this is the following, perhaps it is better to say that one hypothesis might be….similar to what has been seen with…
Author Response
Dear reviewer
We want to thank your careful work and greatly appreciate your comments and suggestions. Based on these comments, we carried out a thorough revision of the manuscript. We believe we fully complied with the proposals. In your opinion, moderate English changes were required; therefore, we used MDPI’s English editing service, so the manuscript was checked by a native English-speaking editor (English Editing ID: 61401). All the modifications were accepted, but our changes are still visible by the “Track change” function. All comments/suggestions and the corresponding responses are listed below, along with the modifications in the revised manuscript, including page and line numbers.
Sincerely yours,
Zsolt Matula
Referee report
This manuscript examines the variability of BCMA expression on the surface of MM patient cells (by establishing cell cultures from 12 primary patients) and its relationship to the ability of a BCMA targeting agent, belantamab mafodotin (belamaf) to effectively kill MM cells either in monoculture or co-cultured with bone marrow stromal cells (BM-MSCs). In an independent series of experiments, they clearly show that mitochondrial transfer (MT) from BM-MSC to MM cells increases with increasing drug concentration. In contrast, MT from MM cells to BM-MSCs decreased with belamaf treatment.
The manuscript is well-written, clear, concise and the data are nicely presented. The manuscript will be of general interest to the MM research community, owing to the fact that belamaf is being used in a number of clinical trials as both monotherapy and in combination with other drugs.
Concepts and small edits to consider including or clarifying:
- Since these cells are from primary patients, the drug resistance is not truly acquired resistance due to continuous drug treatment as in patients being treated. It would be useful if this is considered throughout the manuscript. For instance, in the abstract, consider replacing “MMs acquire noticeable drug resistance in this way” with “MMs are more resistant to belamaf”.
We agree with the suggestion. This is not an acquired drug resistance since these patients have never received belantamab mafodotin treatment earlier. We corrected the corresponding sentences in the abstract, results, discussion, and also in conclusions:
” Here, we show that primary MMs isolated from different patients can vary widely in terms of BCMA expression level, and inadequate expression is associated with extremely high drug resistance to belamaf according to our cytotoxicity assay.” (abstract, page 1, line 25)
” MMs acquire noticeable drug resistance become more resistant to belamaf in this way, similar to other medications we have analyzed earlier in this regard…” (abstract, page 1, line 28)
” Previously, we revealed that conventional anti-myeloma medicines such as the proteasome inhibitor carfilzomib, the BCL-2 inhibitor venetoclax, or the HDAC inhibitor Na-valproate induce an intensive, bidirectional transfer of functional mitochondria between bone marrow stromal cells and MMs, which provides remarkable drug resistance to these pharmaceuticals for the malignant plasma cells.” (results, page 5, line 194)
” The results obtained in the case of belamaf were highly similar: compared to untreated co-cultures, malignant plasma cells received even more functional mitochondria from BM-MSCs in the presence of belamaf due to the treatment, which promoted their drug resistance to belamaf.” (discussion, page 7, line 317)
” However, when cultured with autologous BM-MSCs, the drug resistance of MMs to belamaf increases, although not significantly at all each concentrations. In our co-culture model, the drug resistance and survival of MMs is were significantly enhanced in answer to belamaf treatment by the acquisition of functional mitochondria derived from BM-MSCs in answer to belamaf treatment, similar to the proteasome inhibitor carfilzomib, the BCL-2 inhibitor venetoclax, or the HDAC inhibitor Na-valproate. ” (conclusions, page 8, lines 347-351)
” Here, we show that the same drug resistance mechanism occurs in vitro as a result of belamaf treatment. ” (conclusions, page 8, line 355)
- Page 2, line 94 insert “is” between mab and based
We inserted "is" into the indicated place.
- Page 3, line 117: replace “serve” with “led to”
We replaced “serve” with “led to”.
- Page3, line 119: replace “ growing” with “increasing”
We replaced “growing” with “increasing”.
- Page 5, line 196: replace “conducted” with “led”
We replaced “conducted” with “led”.
- Page 6, line 221: change myeloma multiplex to multiple myeloma
English editor changed "myeloma multiplex" to "it" in the text because of word repetition.
“Although multiple myeloma accounts for approximately 10% of hematologic malignancies and has the second highest incidence, myeloma multiplex it is still generally considered an incurable disease.” (discussion, page 6, line 232)
- On page 7, line 315. Rather than say our explanation for this is the following, perhaps it is better to say that one hypothesis might be….similar to what has been seen with…
We reworded the sentence based on your suggestion (discussion, page 8, line 330):
“One hypothesis explaining this phenomenon might be that the inhibition of actin or tubulin polymerization hinders the transfer of mitochondria through the tunneling nanotubes (TNTs) of plasma cell origin since it has already been demonstrated that MT through TNTs requires both actin and tubulin polymerization. Our explanation for this phenomenon is that the inhibition of actin or tubulin polymerization hinders the transfer of mitochondria through the tunneling nanotubes (TNTs) of plasma cell origin since the MT through TNTs requires both actin and tubulin polymerization. However, when TNT formation and organelle transfer are inhibited through these structures…
